# Relationship of sleep and activity, assessed via a wristwatch-type pulsimeter with an accelerometer, with health status in community-dwelling older adults: A preliminary study

**Akiyo Sasaki-Otomaru** [1,2*], **Kyoko Saito**[3], **Kotaro Yamasue**[4], **Osamu Tochikubo**[4], **Yuka Kanoya**[2]

**1** Research Department, The Dia Foundation for Research on Ageing Societies, Tokyo, Japan, **2** Department of Gerontological Nursing, School of Medicine Nursing Course, Yokohama City University, Yokohama, Japan, **3** Shukutoku University, Iruma, Japan, **4** Department of Public Health, School of Medicine, Yokohama City University, Yokohama, Japan

* sasaki@dia.or.jp

## Abstract

Wearable devices have the potential to promote a healthy lifestyle; however, studies on the use of wearable devices in monitoring health in older adults are limited. We aimed to investigate the relationship of sleep and activity data with health status among older adults. Fifty-five community-dwelling older adults were asked to wear a wristwatch-type wearable device (the Pulsense [PS]) and measure home blood pressure (HBP) over a period of 5–7 consecutive days. Deep-sleep duration, physical and mental activity duration, and body-movement duration were obtained from PS data using special software. We also collected data on demographics and physical and mental health status. We found that the body-movement duration in women was longer than that in men. Among men, body-movement duration was strongly and negatively correlated with the Kihon Checklist (KCL) score. It also showed moderate correlations with the Geriatric Depression score, physical functioning, bodily pain, vitality, social function, and role emotional scores from the Medical Outcomes Survey Short Form-8 questionnaire, as well as with hand-grip strength. There was no significant correlation between monitoring data and health status in women. In the multiple linear regression analysis, body-movement duration was negatively associated with age and the KCL score. KCL is a common questionnaire for screening frailty in Japan. Our results showed that body-movement duration was negatively associated with age and the KCL score, suggesting the potential of PS in guiding personalized health management of older community-dwelling adults with risks of frailty.

## Introduction

Wearable devices have become a topic of interest in recent years [1]. These devices can measure vital signs such as heart rate, respiration rate, and body temperature, in addition to physical

**Data availability statement:** This study has sensitive participant information and cannot be released publicly due to ethical and legal restrictions imposed by the Ethical Committee of Yokohama City University School of Medicine. All data supporting the findings of this study are available upon reasonable request to the Department of Medical Education, Yokohama City University School of Medicine (Email: inamorim@yokohama-cu.ac.jp).

**Funding:** This work was supported by the 31st Foundation for Total Health Promotion and Grant-in-Aid for Young Scientists (B), under Grant number. 17K17508(KAKENHI). The funders had no role in study design, data collection and analysis, decision to publish, or preparation of the manuscript.

**Competing interests:** The authors have declared that no competing interests exist.

activity and sleep conditions. Accelerometers, which were once mainly worn on the waist, are increasingly being replaced by wrist-type models [2,3]. An accelerometer could evaluate not only moderate-to-vigorous physical activity but also low physical activity (LPA) and sedentary behavior; LPA accounted for a large proportion of total physical activity time among older adults with declining functionality [4]. The Canadian 24-Hour Movement Guideline provides evidence-based recommendations for a healthy day (24-h), comprising a combination of sleep, sedentary behaviors, and light-intensity and moderate-to-vigorous-intensity physical activity [5]. The wristwatch-type wearable devices are expected to provide 24-h monitoring.

We previously used a wristwatch-type pulsimeter with accelerometer (PS-500B; Pulsense®, Seiko Epson Co. Ltd., Nagano, Japan), which is an improvement and productization of previously reported models [6–8]. Our study showed that home blood pressure (HBP) was affected by sleep, even after adjusting for age and body mass index (BMI) among healthy adults [8]. Therefore, we considered that wearable devices can be used to comprehensively assess the lifestyle of older adults.

Japan has the world's highest life expectancy and a persistently low birth rate. Japan's population is aging more rapidly than that of any other country [9]. With the increasing number of older adults, a substantial increase in health care and social security costs is expected. The ratio of individuals in need of nursing care increases because of dementia and physical functional deterioration. In Japan, the main reasons for needing nursing care are dementia, cerebrovascular disease, and frailty [10]; therefore, healthy behaviors, such as increased physical activity and improved quality of sleep, are important factors in preventing these diseases and symptoms.

The use of wristwatch-type devices has the potential to promote healthy lifestyles in terms of physical activity and weight [11,12]. However, studies on the use of wearable devices in monitoring health in older adults are limited [13,14]. Data on sleep and activity duration in healthy older individuals would be valuable for monitoring purposes. This study aimed to examine the relationship of sleep and activity data (obtained using Pulsense) with the health status of community-dwelling older adults.

## Materials and methods

### Participants

From July 2015 to August 2017 and from June 2020 to October 2020, we recruited community-dwelling older adults participating in health classes at local health centers. The eligibility criteria were as described in the previous study [8]. Of the 61 participants in the present study, six were excluded due to a lack of sleep and activity data. Finally, we included 55 older adults.

### Ethical considerations

The research was conducted according to the principles of the World Medical Association Declaration of Helsinki. Informed consent was obtained from all participants, and this cross-sectional study was approved by the Ethical Committee of the Yokohama City University School of Medicine (approval numbers: A150326011 and A200200005).

### Variables

**Sleep and activity.** Sleep and activity data were measured using the Pulsense (henceforth termed PS) and processed using special software to yield data on deep-sleep, physical-activity, mental- activity, and body-movement durations [8]. The participants wore the PS over a period of 5–7 consecutive days and nights, as described in a previous study [8].

**HBP.** HBP was measured twice a day, in the morning and at night, in a sitting position after at least 10 min of rest according to the guidelines, and the data were averaged over

a 5- to 7-day period for each participant [15]. Hypertension was defined as the use of antihypertensive drugs, systolic blood pressure (SBP) ≥ 135 mmHg, and/or diastolic blood pressure (DBP) ≥ 85 mmHg.

**Kihon checklist and depression.** The Kihon Checklist (KCL) score was used for a comprehensive evaluation of the physical, psychological, functional, and social status of the participants. The KCL consists of five subdomains (physical, nutritional, oral, cognitive function, and depressive mood). Each KCL score indicates the perceived level of difficulty of the activity in the question, and a higher score on the checklist indicates a higher risk of requiring support in each domain. A total score of ≥6 indicated frailty [16]. The current study applied the KCL without the depression score [17]. Instead, the Geriatric Depression Test-5 (GDS-5) was used to evaluate depressive mood, according to a previous study [16]. A higher score on the GDS indicated a higher risk of depression [18].

**Subjective sleep.** Subjective sleep was assessed using the Pittsburgh Sleep Quality Index (PSQI), which included 19 self-administered questions that covered seven domains, namely subjective quality, sleep latency, sleep duration, habitual sleep efficiency, sleep disturbance, sleep medication use, and daytime dysfunction. Sleep disturbance was defined as having a PSQI score ≥ 6 [19].

**Health-related quality of life.** Health-related quality of life was assessed by administering the Medical Outcomes Survey Short Form-8 questionnaire (SF-8) [20]. The SF-8 consisted of eight items, each representing one health profile dimension: general health perception, physical functioning, role functioning–physical, bodily pain, vitality, social functioning, mental health, and role functioning–emotional. The Japanese version of the SF-8 met the standard criteria for content, construct, and criterion validity, as it was based on a nationwide survey [20].

**Physical function.** Measures of physical function included gait speed (in m/s) and hand-grip strength (in kg) and were assessed to reflect physical functional status. To evaluate gait speed, the participants were asked to walk 11 m at their usual pace, and the time taken to traverse the middle 5 m of this distance was measured once [21]. The recorded times were converted to speed (in m/s). Hand-grip strength (kg) was measured using a digital hand dynamometer (T.K.K.5401; Takei Scientific Instruments Co., Ltd., Niigata, Japan). The grip strength of each hand was assessed twice, and the maximal value of each hand was averaged for analysis.

**Statistical analysis.** All data were analyzed using SPSS Statistics version 28 (IBM Corp., Chicago, IL, USA). The normality of the data distribution was tested using the Shapiro–Wilk test. The KCL score, PSQI score, GDS score, physical activity duration, and mental activity duration data were not normally distributed. Therefore, differences between the sexes were compared using the Mann–Whitney U-test. Spearman's correlation test was used to assess the relationship between the PS data and health status measurements, including HBP. We interpreted the relative strength of correlation as follows: < 0.1 = "negligible," 0.1–0.39 = "weak," 0.4–0.69 = "moderate," 0.7–0.89 = "strong," and 0.9–1.0 = "very strong" [22]. Lastly, multiple linear regression analyses were used to explore the association between the PS data (deep-sleep and body-movement durations) and variables identified as significant by the chi-square test and Spearman's correlation test. All analyses were two-sided at the 5% level of significance.

## Results

### Participant characteristics

Of the 55 participants, 35 (63.6%) were women and 26 (47.3%) had hypertension. The mean age of the participants was 73.8 years. Table 1 compares the characteristics of the participants according to sex. There were significant differences in age, PSQI score, hand-grip strength, and body-movement duration between men and women.

**Table 1. Characteristics of participants.**

| Parameters | Men | | | | | Women | | | | | |
|---|---|---|---|---|---|---|---|---|---|---|---|
| | n | mean | median | min | max | n | mean | median | min | max | p |
| Age (years) | 20 | 76.0 | 76.0 | 67.0 | 91.0 | 35 | 72.6 | 73.0 | 65.0 | 82.0 | 0.032* |
| BMI (kg/m2) | 20 | 23.2 | 22.9 | 17.7 | 31.2 | 35 | 21.7 | 21.8 | 15.5 | 32.8 | 0.144 |
| Living status (single; n, %) | | 1 | (11.1) | | | | 8 | (88.9) | | | 0.085 |
| Kihon Checklist (Total score; 1–20) | 20 | 3.4 | 2.5 | 0.0 | 12.0 | 35 | 3.7 | 4.0 | 0.0 | 8.0 | 0.457 |
| Kihon Checklist (6 or more; n, %) | | 3 | (15.0) | | | | 8 | (22.9) | | | 0.483 |
| GDS-5 | 20 | 0.9 | 0.0 | 0.0 | 4.0 | 35 | 0.6 | 0.0 | 0.0 | 3.0 | 0.814 |
| Hypertension (n, %) | | 11 | (55.0) | | | | 15 | (45.5) | | | 0.279 |
| Use of antihypertensive drugs (n, %) | | 7 | (35.0) | | | | 10 | (28.6) | | | 0.420 |
| SF-8 Physical functioning | 20 | 47.89 | 50.27 | 34.38 | 63.38 | 35 | 49.79 | 50.27 | 26.89 | 63.38 | 0.366 |
| Role physical | 20 | 48.53 | 53.54 | 16.69 | 53.54 | 35 | 48.37 | 47.77 | 27.59 | 53.54 | 0.133 |
| Bodily pain | 20 | 50.46 | 54.09 | 21.80 | 54.09 | 35 | 50.09 | 54.09 | 27.91 | 54.09 | 0.378 |
| General health | 20 | 50.78 | 52.46 | 38.21 | 60.35 | 35 | 48.28 | 46.10 | 38.21 | 60.35 | 0.174 |
| Vitality | 20 | 49.78 | 53.74 | 28.68 | 60.01 | 35 | 50.85 | 53.74 | 38.51 | 60.01 | 0.840 |
| Social functioning | 20 | 51.08 | 55.14 | 26.00 | 55.14 | 35 | 51.69 | 55.14 | 37.65 | 55.14 | 0.347 |
| Role emotional | 20 | 51.95 | 53.83 | 36.30 | 56.93 | 35 | 51.81 | 50.72 | 36.30 | 56.93 | 0.800 |
| Mental health | 20 | 51.24 | 54.19 | 31.42 | 54.19 | 35 | 51.41 | 54.19 | 42.24 | 54.19 | 0.540 |
| PSQI | 20 | 4.5 | 4 | 0 | 11 | 35 | 7.4 | 7 | 0 | 15 | 0.009* |
| PSQI scores (6 or more; n, %) | | 7 | (35.0) | | | | 20 | (57.1) | | | 0.114 |
| Hand-grip strength (kg) | 20 | 33.6 | 34.9 | 18.0 | 45.8 | 35 | 21.9 | 21.3 | 15.5 | 32.3 | <0.001* |
| Walking-speed (m/s) | 19 | 1.5 | 1.4 | 0.6 | 2.5 | 35 | 1.5 | 1.5 | 1.0 | 2.0 | 0.458 |
| PS Deep-sleep duration (h) | 20 | 4.26 | 3.98 | 1.76 | 7.75 | 35 | 4.97 | 5.00 | 2.06 | 8.92 | 0.080 |
| Physical activity duration (h) | 20 | 2.17 | 2.09 | 0.14 | 5.77 | 35 | 2.36 | 2.21 | 0.35 | 7.63 | 0.643 |
| Mental activity duration (h) | 20 | 0.99 | 0.66 | 0.08 | 3.81 | 35 | 0.74 | 0.62 | 0.04 | 3.10 | 0.489 |
| Body-movement duration (h) | 20 | 6.01 | 5.57 | 3.42 | 8.89 | 35 | 7.09 | 7.15 | 4.73 | 9.81 | 0.026* |

PSQI, Pittsburgh Sleep Quality Index; PS, a pulsimeter (Pulsense®); GDS-5, Geriatric Depression Test-5; SF-8, Medical Outcomes Survey Short Form-8 questionnaire.

Mann–Whitney U test, Chi-square test.

*p < 0.05.

## PS data and health status

Spearman's rank correlation coefficients for correlations of sleep and activity data with other variables in each sex are shown in Tables 2 and 3. Among male participants, a strong negative correlation was found between physical activity duration and KCL score ($rs = −0.793$, $p < 0.001$) and a moderate negative correlation with GDS-5 score ($rs = −0.463$, $p = 0.040$). Physical activity duration showed moderate positive correlations with physical functioning ($rs = 0.652$, $p = 0.002$), role physical ($rs = 0.511$, $p = 0.021$), bodily pain ($rs = 0.511$, $p = 0.021$), vitality ($rs = 0.635$, $p = 0.003$), and social functioning ($rs = 0.568$, $p = 0.009$) in the SF-8. Physical activity duration was also moderately correlated with hand-grip strength ($rs = 0.462$, $p = 0.040$) and walking-speed ($rs = 0.460$, $p = 0.048$). Body-movement duration showed strong correlations with KCL score and physical functioning in SF-8. Additionally, deep-sleep duration was moderately correlated with BMI in male participants.

In contrast, no significant correlation was found between PS data and health status among female participants.

**Table 2. Spearman's rank correlation coefficients between sleep and activity data and variables in male participants (n = 20).**

| | | Deep-sleep duration | Physical activity duration | Mental activity duration | Body- movement duration |
|---|---|---|---|---|---|
| **Age** | rs | −0.200 | −0.187 | −0.077 | −0.280 |
| | p | 0.398 | 0.430 | 0.747 | 0.231 |
| **BMI** | rs | 0.522 | −0.166 | −0.275 | −0.215 |
| | p | 0.018* | 0.484 | 0.241 | 0.362 |
| **Kihon Checklist score** | rs | −0.068 | −0.793 | −0.311 | −0.701 |
| | p | 0.775 | <0.001* | 0.182 | <0.001* |
| **GDS-5** | rs | 0.203 | −0.463 | −0.011 | −0.538 |
| | p | 0.392 | 0.040* | 0.963 | 0.014* |
| **SF-8 Physical functioning** | rs | 0.004 | 0.652 | 0.126 | 0.779 |
| | p | 0.986 | 0.002* | 0.598 | <0.001* |
| **Role physical** | rs | −0.182 | 0.511 | 0.236 | 0.469 |
| | p | 0.442 | 0.021* | 0.315 | 0.037* |
| **Bodily pain** | rs | −0.182 | 0.511 | 0.236 | 0.469 |
| | p | 0.442 | 0.021* | 0.315 | 0.037* |
| **General health** | rs | 0.230 | 0.206 | 0.240 | 0.210 |
| | p | 0.329 | 0.384 | 0.307 | 0.375 |
| **Vitality** | rs | −0.083 | 0.635 | 0.239 | 0.610 |
| | p | 0.729 | 0.003* | 0.310 | 0.004* |
| **Social functioning** | rs | −0.223 | 0.568 | 0.189 | 0.522 |
| | p | 0.345 | 0.009* | 0.425 | 0.018* |
| **Role emotional** | rs | −0.069 | 0.348 | −0.019 | 0.473 |
| | p | 0.772 | 0.133 | 0.938 | 0.035* |
| **Mental health** | rs | −0.148 | 0.443 | 0.169 | 0.385 |
| | p | 0.532 | 0.050 | 0.476 | 0.094 |
| **PSQI** | rs | 0.179 | −0.019 | 0.171 | −0.114 |
| | p | 0.449 | 0.937 | 0.471 | 0.634 |
| **Hand-grip strength** | rs | −0.263 | 0.462 | −0.038 | 0.614 |
| | p | 0.262 | 0.040* | 0.875 | 0.004* |
| **Walking-speed** | rs | −0.125 | 0.460 | 0.167 | 0.465 |
| **(n = 19)** | p | 0.611 | 0.048* | 0.495 | 0.045* |

GDS-5, Geriatric Depression Test-5; PSQI, Pittsburgh Sleep Quality Index; SF-8, Medical Outcomes Survey Short Form-8 questionnaire.

*p < 0.05.

## Association between body-movement duration and health status

Multilinear regression analyses were conducted to explore the associations between demographic factors and body-movement duration. BMI, hypertension, PSQI, and KCL were selected as predictor variables, with age and gender included as covariates. As shown in Table 4, the KCL score was negatively associated with body-movement duration (β=−0.437, p < 0.001) after adjusting for age and gender. No significant model could be established for factors influencing deep-sleep duration after multiple regression analysis for deep-sleep duration and independent variables.

**Table 3. Spearman's rank correlation coefficients between sleep and activity data and variables in female participants (n = 35).**

| | | Deep-sleep duration | Physical activity duration | Mental activity duration | Body- movement duration |
|---|---|---|---|---|---|
| **Age** | rs | 0.284 | −0.289 | −0.257 | −0.254 |
| | p | 0.098 | 0.093 | 0.135 | 0.141 |
| **BMI** | rs | −0.040 | −0.319 | −0.317 | −0.080 |
| | p | 0.820 | 0.062 | 0.063 | 0.650 |
| **Kihon Checklist score** | rs | −0.062 | −0.017 | 0.149 | −0.284 |
| | p | 0.725 | 0.925 | 0.393 | 0.098 |
| **GDS-5** | rs | −0.003 | −0.108 | −0.015 | 0.028 |
| | p | 0.987 | 0.535 | 0.931 | 0.874 |
| **SF-8 Physical functioning** | rs | 0.222 | −0.211 | −0.207 | <0.001 |
| | p | 0.200 | 0.223 | 0.233 | 0.999 |
| **Role physical** | rs | 0.079 | −0.127 | −0.198 | 0.080 |
| | p | 0.653 | 0.466 | 0.253 | 0.649 |
| **Bodily pain** | rs | 0.009 | −0.072 | −0.080 | −0.094 |
| | p | 0.958 | 0.682 | 0.647 | 0.591 |
| **General health** | rs | 0.151 | −0.105 | 0.010 | −0.127 |
| | p | 0.386 | 0.548 | 0.953 | 0.469 |
| **Vitality** | rs | 0.133 | −0.177 | −0.188 | 0.012 |
| | p | 0.446 | 0.310 | 0.281 | 0.947 |
| **Social functioning** | rs | 0.091 | −0.079 | −0.160 | 0.059 |
| | p | 0.603 | 0.653 | 0.358 | 0.737 |
| **Role emotional** | rs | 0.056 | −0.017 | −0.122 | 0.090 |
| | p | 0.747 | 0.923 | 0.486 | 0.607 |
| **Mental health** | rs | −0.058 | 0.053 | −0.081 | 0.101 |
| | p | 0.740 | 0.761 | 0.642 | 0.563 |
| **PSQI** | rs | −0.029 | 0.142 | 0.279 | −0.084 |
| | p | 0.868 | 0.417 | 0.104 | 0.632 |
| **Hand-grip strength** | rs | 0.129 | −0.130 | −0.106 | −0.042 |
| | p | 0.461 | 0.457 | 0.545 | 0.812 |
| **Walking-speed** | rs | 0.114 | 0.101 | −0.054 | 0.320 |
| | p | 0.516 | 0.565 | 0.760 | 0.061 |

GDS-5, Geriatric Depression Test-5; PSQI, Pittsburgh Sleep Quality Index; SF-8, Medical Outcomes Survey Short Form-8 questionnaire.

*p < 0.05.

**Table 4. Multilinear regression analysis between body-movement duration and independent variables (n = 55).**

| | β | 95% CI | | p |
|---|---|---|---|---|
| **Age** | −0.274 | −0.15 | −0.008 | 0.040* |
| **BMI** | −0.120 | −0.154 | 0.055 | 0.347 |
| **KCL** | −0.437 | −0.409 | −0.124 | <0.001* |
| **PSQI scores** | −0.026 | −0.105 | 0.085 | 0.832 |
| **Hypertension** | −0.113 | −1.116 | 0.417 | 0.364 |
| **Sex (men = 1)** | −0.250 | −1.642 | 0.036 | 0.060 |
| **Adjusted-R²** | 0.370 | | | |
| **p** | <0.001 | | | |

BMI, body mass index; KCL, Kihon Checklist; PSQI, Pittsburgh Sleep Quality Index; CI, confidence interval.

*p < 0.05.

## Discussion

The present study used physical and sleep data from PS to clarify their relationship with health status. Our results showed that shorter body-movement duration was associated with a higher risk of frailty in older adults after adjusting for age and sex. Frailty prevention is an important topic, particularly concerning Japanese older adult women [23]. The KCL is a governmental standardized index assessing frailty in Japanese individuals [24].

The body-movement duration in women was longer than that in men. The body-movement duration assessed by the PS included light physical activity, such as housework and cooking [25]. A nationwide survey in Japan reported that older women spent more time on housework compared to older men [26]. Another previous study showed that women spent more time in LPA compared to men [4]. Additionally, a cross-sectional study showed that light physical activity was positively associated with physical performance [27]. Light physical activity indicates 2.0–2.9 metabolic equivalents (METs) of physical activity, which includes cooking, feeding household animals, or doing laundry [28]. These activities might be easy to adopt into the lifestyle of older adults of both sexes. Although body-movement duration by PS did not imply LPA, PS might be a useful evaluator of light physical activity to set a target for maintaining and/or increasing body-movement duration. This is so that older adults can understand their own activity time and adjust it as needed.

PS also evaluates physical and mental activities. In this study, physical and mental activity data were not normally distributed; therefore, we did not conduct a multiple linear regression analysis. In simple linear regression analysis, physical activity duration correlated negatively with the KCL and GDS scores and positively with several items of the SF-8. Both physical and mental activity durations might be associated with health status; however, further research is needed to determine the specific factors that influence activity duration. Sedentary behavior in older adults has been linked with negative health outcomes [29]; older people are not always able to exercise because of underlying health conditions. However, PS evaluates acceleration and pulse. Future research should explore the duration of mental activity using PS to identify favorable sedentary behavior patterns.

Deep-sleep duration correlated positively with BMI in simple linear regression in men. Although short sleep duration is generally associated with an increased risk of obesity [30,31], a study reported that long sleep duration was associated with sarcopenic obesity in older men [32]. In our study, we did not find a significant model when deep-sleep duration was used as the dependent variable in a multiple regression analysis. This suggested that further investigation was needed.

In addition, a gender-specific approach is necessary to maintain and improve activity duration. Men spend their time on various activities, such as work, housework, and hobbies. Further reducing these activity times can have negative effects on their physical and mental health. Therefore, it is important to avoid further reducing time to maintain activity levels. Women have more reasons for decreased activity levels than men, such as injuries from falls and fractures. It is important to examine these reasons individually and take appropriate measures. For older adults, it is particularly important for family and caregivers to cooperate in monitoring physical activity duration. Family members living far away can check on the older adult's condition by phone or video call and encourage them to exercise as needed. Caregivers can also observe and record the older adult's activity time in their daily lives.

Our study had some limitations. First, a major methodological limitation was the inability to determine causality. Second, only a small number of participants were included. To detect a moderate correlation of 0.4 using R software, we estimated a required sample size of 47 and aimed to recruit 50 participants. Although we successfully recruited 55 individuals, our gender-stratified analysis resulted in smaller sample sizes within each subgroup.

Moreover, participants were generally health-conscious, which may limit generalizability. Third, although sleep and activity durations may be affected by the season of the year, the data were collected over periods throughout the year due to weather considerations. This resulted in a survey that was not a comprehensive daily 1-week assessment. Finally, we did not collect information on the nutritional status, which could notably influence frailty. Lastly, the PS could measure only activity and sleep. Other wearable devices with oximeters, electrocardiograms, and other functions are also currently available and warrant further research.

## Conclusions

This cross-sectional study suggests that decreased body-movement duration may indicate a potential risk for frailty in older adults. Monitoring physical activity duration and body-movement duration by PS could be important data not only for older adults themselves but also for their families, especially those living far away, and informal caregivers who cannot observe them constantly.

## Acknowledgments

We would like to thank all participants and the staff of the local health centers. We also extend our gratitude to Ms. Kaori Yoshida from the Gerontological Nursing Department at Yokohama City University for her assistance in data collection and Dr Masahiko Inamori from the Department of Medical Education at Yokohama City University for his contributions in data management.

## Author contributions

**Conceptualization:** Akiyo Sasaki-Otomaru, Kotaro Yamasue.

**Data curation:** Akiyo Sasaki-Otomaru, Kyoko Saito.

**Formal analysis:** Akiyo Sasaki-Otomaru, Kyoko Saito.

**Funding acquisition:** Akiyo Sasaki-Otomaru.

**Investigation:** Akiyo Sasaki-Otomaru, Kyoko Saito.

**Methodology:** Akiyo Sasaki-Otomaru, Kotaro Yamasue, Osamu Tochikubo.

**Project administration:** Akiyo Sasaki-Otomaru.

**Resources:** Akiyo Sasaki-Otomaru.

**Software:** Kotaro Yamasue, Osamu Tochikubo.

**Supervision:** Osamu Tochikubo, Yuka Kanoya.

**Validation:** Akiyo Sasaki-Otomaru, Kyoko Saito.

**Visualization:** Akiyo Sasaki-Otomaru.

**Writing – original draft:** Akiyo Sasaki-Otomaru.

**Writing – review & editing:** Akiyo Sasaki-Otomaru, Kyoko Saito, Kotaro Yamasue, Osamu Tochikubo, Yuka Kanoya.

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
