## [Decision Letter · Decision Letter 0]

2 Oct 2024

PONE-D-24-27156Relationship of sleep and activity, assessed via a wrist-watch-type pulsimeter with an accelerometer, with health status in community-dwelling older adults: A preliminary studyPLOS ONE

Dear Dr. Sasaki-Otomaru,

Thank you for submitting your manuscript to PLOS ONE. After careful consideration, we feel that it has merit but does not fully meet PLOS ONE’s publication criteria as it currently stands. Therefore, we invite you to submit a revised version of the manuscript that addresses the points raised during the review process.

**ACADEMIC EDITOR:**

We look forward to receiving your revised manuscript.

Kind regards,

Hidetaka Hamasaki

Academic Editor

PLOS ONE

Journal Requirements:

2. Thank you for stating the following financial disclosure: This work was supported by the 31st Foundation for Total Health Promotion and the Grant-in-Aid for Young Scientists (B), under Grant number. 17K17508 (KAKENHI).

3. In the online submission form, you indicated that the dataset generated and/or analyzed during the current study are not publicly due to ethical considerations. The data that support the findings of this study are available from the corresponding author upon reasonable request. 

Reviewers' comments:

Reviewer's Responses to Questions

**Comments to the Author**

1. Is the manuscript technically sound, and do the data support the conclusions?

Reviewer #1: Yes

Reviewer #2: Partly

2. Has the statistical analysis been performed appropriately and rigorously? 

Reviewer #1: Yes

Reviewer #2: Yes

3. Have the authors made all data underlying the findings in their manuscript fully available?

Reviewer #1: Yes

Reviewer #2: Yes

4. Is the manuscript presented in an intelligible fashion and written in standard English?

Reviewer #1: Yes

Reviewer #2: Yes

5. Review Comments to the Author

Reviewer #1: The purpose of this descriptive study was to examine the relationship between sleep and activity data and health status among older adults. It appears that this research fills some gaps in the literature. The introduction concluded with an explicit purpose statement. The methods are complete except I would like you to find and use a scale to qualify the magnitude of associations. Someone could replicate this study based on the methods section. The results, discussion, and conclusion are consistent with the primary purpose. However, again, I would like to see more in the discussion about the limits of correlation analyses. Specific comments are made in the pdf.

Reviewer #2: The title "Association of home blood pressure with sleep and physical and mental activity, assessed via a wristwatch-type pulsimeter with accelerometer in adults" is not compatible whith the content of this manuscript.

How the sample size was calculated?

What is the main research problem and what are the hypotheses? Dependent and independent variables need to be established.

Why the authors did not collect information on the nutritional status, which could notably influence frailty.

In male participants, physical activity duration correlated positively with bodily pain. Could the authors explain that?

Multilinear regression analyses were conducted to explore the associations between demographic factors and body-movement duration. Why the autors chose so kind of variables for this analises?

The results showed that body movement duration decreased in correlation with the higher risk of frailty in older adults after adjusting for sex. I dont understand this conclusion and it is not supraising probably.

The conclusions are over-interpretation.

6. PLOS authors have the option to publish the peer review history of their article (what does this mean? ). If published, this will include your full peer review and any attached files.

**Do you want your identity to be public for this peer review?** For information about this choice, including consent withdrawal, please see our Privacy Policy .

Reviewer #1: No

Reviewer #2: No

---

## [Author Response · Author response to Decision Letter 1]

10 Dec 2024

Dear Dr. Hidetaka Hamasaki,

Thank you for the thoughtful and constructive feedback on our manuscript, “Relationship of sleep and activity, assessed via a wristwatch-type pulsimeter with an accelerometer, with health status in community-dwelling older adults: A preliminary study.” We have addressed the reviewer’s comments, and the revised sections are highlighted in yellow.

We hereby resubmit our manuscript for a secondary evaluation. Thank you again for considering our paper.

Sincerely,

Akiyo SASAKI-OTOMRU

Journal Requirements:

Response

Thank you for your valuable comment. We have revised our manuscript to comply with PLoS ONE’s style requirements, including to the title page, reference format, and placement of Table 2.

2. Thank you for stating the following financial disclosure: This work was supported by the 31st Foundation for Total Health Promotion and the Grant-in-Aid for Young Scientists (B), under Grant number. 17K17508 (KAKENHI).

Response

Thank you for your insightful comment. In our study, the funders had no role. We have stated in our cover letter: "The funders had no role in study design, data collection and analysis, decision to publish, or preparation of the manuscript."

3. In the online submission form, you indicated that the dataset generated and/or analyzed during the current study are not publicly due to ethical considerations. The data that support the findings of this study are available from the corresponding author upon reasonable request.

Response

Thank you for your perceptive comment. This study has sensitive participant information and cannot be released publicly due to ethical and legal restrictions imposed by the Ethical Committee of Yokohama City University School of Medicine. All data supporting the findings of this study are available upon reasonable request to the Department of Medical Education (inamorim@yokohama-cu.ac.jp), Yokohama City University School of Medicine. This information is also described in the online submission form.

4. Please review your reference list to ensure that it is complete and correct. If you have cited papers that have been retracted, please include the rationale for doing so in the manuscript text, or remove these references and replace them with relevant current references. Any changes to the reference list should be mentioned in the rebuttal letter that accompanies your revised manuscript. If you need to cite a retracted article, indicate the article’s retracted status in the References list and also include a citation and full reference for the retraction notice.　

Response

Thank you for your important comment. We have replaced the reference to Jakicic et al. (2016) with Zhou et al. (2020) for reference number 12. Additionally, we have corrected the DOI for reference number 24, which is now listed as number 25 in the revised manuscript.

We have revised males and females to men and women in Table 1. Furthermore, we have used “deep-sleep duration” and “body-movement duration” with hyphen to maintain consistency.

Reviewers' comments:

Reviewer #1:

Please provide a scale that describes the relative strength of correlation.

Response

Thank you for your meaningful comment. We have added the relative strength of correlation in the Statistical analysis section (lines 137-139). Following the addition of reference number 22 in the revised manuscript, the subsequent reference numbers have been updated accordingly.

Some items in the table are not aligned correctly which may simply be a function of the size of the table.

Response

Thank you for pointing this out. We have corrected the row formatting in Table 1.

As noted earlier you have some statistically significant correlations that may not be that clinically meaningful.

Response

Thank you for your thoughtful comment. We have added the interpretation of correlations in the Results section (lines 155-163).　

“Strongest” but still may not be all that meaningful clinically.

Response

Thank you for bringing this to our attention. We have revised the relevant part in the Results section (lines 177-179).

Prior to restarting the purpose of the study please also restate the main problem you are attempting to address and the need for this study

Response

We appreciate your attention to this matter. We have added the main problem and our hypothesis at the beginning of the Discussion section (lines 186-187)

This statement “decreased in correlation” is too general/broad/vague.

Response

Your meaningful comment is duly noted. We have revised the text as suggested (lines 187-188).

A major methodological limitation is that correlation and regression only looks at relationships and not cause/effect

Response

Your insightful comment is recognized. We have revised the limitations paragraph in the Discussion section in response to your comment (lines 233-241).

Reviewer #2: The title "Association of home blood pressure with sleep and physical and mental activity, assessed via a wristwatch-type pulsimeter with accelerometer in adults" is not compatible whith the content of this manuscript.

Response

Thank you for highlighting the title. Our previous research focused on blood pressure; however, the present manuscript examines the relationship between sleep and activity data, assessed using a wristwatch, and health status. Therefore, we have titled it: “Relationship of sleep and activity, assessed via a wristwatch-type pulsimeter with accelerometer, with health status in community-dwelling older adults: A preliminary study.”

How the sample size was calculated?

Response

Your insightful comment is appreciated. We have added an explanation of the power analysis for sample size calculation in the Discussion section (lines 235-238).

What is the main research problem and what are the hypotheses? Dependent and independent variables need to be established.

Response

We appreciate your valuable question. To examine the nature of sleep-activity patterns in the older population, we utilized PS as the dependent variable. Independent variables were selected based on an extensive body of literature exploring factors influencing sleep and activity. We have added our hypothesis in the Introduction section (lines 69-71).

Why the authors did not collect information on the nutritional status, which could notably influence frailty.

Response

We agree with your suggestion and have already included this information in the limitations section (lines 242-243).

At the beginning of the study, we assessed nutritional status using the BDHQ (Brief-Type Self-Administered Diet History Questionnaire). However, because of the substantial burden of the BDHQ’s 80-item questionnaire on participants, and given that initial results indicated good nutritional status, we decided to discontinue its use.

In male participants, physical activity duration correlated positively with bodily pain. Could the authors explain that?

Response

We appreciate your meaningful question. A higher bodily pain score indicates that activity is not interrupted by pain; therefore, we believe our results are consistent.

Multilinear regression analyses were conducted to explore the associations between demographic factors and body-movement duration. Why the autors chose so kind of variables for this analises?

Response

You have raised an important question. Because of the limited number of subjects, we selected five to six variables. Based on previous research, we chose variables expected to be associated with activity duration. Therefore, we selected BMI, hypertension, PSQI, and KCL as predictor variables, with age and gender as covariates. These points have been added (lines 176-177).

The results showed that body movement duration decreased in correlation with the higher risk of frailty in older adults after adjusting for sex. I dont understand this conclusion and it is not supraising probably.

The conclusions are over-interpretation.

Response

Thank you for pointing this out. By clearly stating our hypothesis and revising the limitations, we believe our manuscript is now more understandable. In addition, we have revised the Conclusions section to avoid over-interpretation (lines 248-249). We hope you understand our rationale for these decisions.

---

## [Decision Letter · Decision Letter 1]

30 Dec 2024

Relationship of sleep and activity, assessed via a wrist-watch-type pulsimeter with an accelerometer, with health status in community-dwelling older adults: A preliminary study

PONE-D-24-27156R1

Dear Dr. Sasaki-Otomaru,

We’re pleased to inform you that your manuscript has been judged scientifically suitable for publication and will be formally accepted for publication once it meets all outstanding technical requirements.

Kind regards,

Hidetaka Hamasaki

Academic Editor

PLOS ONE

Additional Editor Comments (optional):

Reviewers' comments:

Reviewer's Responses to Questions

**Comments to the Author**

1. If the authors have adequately addressed your comments raised in a previous round of review and you feel that this manuscript is now acceptable for publication, you may indicate that here to bypass the “Comments to the Author” section, enter your conflict of interest statement in the “Confidential to Editor” section, and submit your "Accept" recommendation.

Reviewer #1: All comments have been addressed

2. Is the manuscript technically sound, and do the data support the conclusions?

Reviewer #1: Yes

3. Has the statistical analysis been performed appropriately and rigorously? 

Reviewer #1: Yes

4. Have the authors made all data underlying the findings in their manuscript fully available?

Reviewer #1: Yes

5. Is the manuscript presented in an intelligible fashion and written in standard English?

Reviewer #1: Yes

6. Review Comments to the Author

Reviewer #1: My earlier questions/comments/suggestions were addressed. I do not have any further questions or comments.

7. PLOS authors have the option to publish the peer review history of their article (what does this mean? ). If published, this will include your full peer review and any attached files.

**Do you want your identity to be public for this peer review?** For information about this choice, including consent withdrawal, please see our Privacy Policy .

Reviewer #1: No

---

## [Editor Report · Acceptance letter]

PONE-D-24-27156R1

PLOS ONE

Dear Dr. Sasaki-Otomaru,

I'm pleased to inform you that your manuscript has been deemed suitable for publication in PLOS ONE. Congratulations! Your manuscript is now being handed over to our production team.

Kind regards,

on behalf of

Dr. Hidetaka Hamasaki

Academic Editor

PLOS ONE